# CALIBRATION OF ORDINAL REGRESSION NETWORKS

## ABSTRACT

Recent studies have shown that deep neural networks are not well-calibrated and produce over-confident predictions. The miscalibration issue primarily stems from the minimization of cross-entropy, which aims to align predicted softmax probabilities with one-hot labels. In ordinal regression tasks, this problem is compounded by an additional challenge: the expectation that softmax probabilities should exhibit unimodal distribution is not met with cross-entropy. Rather, the ordinal regression literature has focused on unimodality and overlooked calibration. To address these issues, we propose a novel loss function that introduces order-aware calibration, ensuring that prediction confidence adheres to ordinal relationships between classes. It incorporates soft ordinal encoding and label-smoothing-based regularization to enforce both calibration and unimodality. Extensive experiments across three popular ordinal regression benchmarks demonstrate that our approach achieves state-of-the-art calibration without compromising accuracy.

## 1 INTRODUCTION

Despite significant advances in ordinal regression tasks, such as medical diagnosis and age estimation, one critical aspect has often been overlooked: calibration. While research has predominantly focused on improving accuracy, the importance of well-calibrated predictions in ordinal regression remains underexplored. In high-risk applications, where both accuracy and reliability are crucial, poorly calibrated models can lead to overconfident or underconfident decisions, potentially resulting in harmful outcomes.

Ordinal regression, also called ordinal classification, involves a natural ordering between class labels, setting it apart from nominal tasks. Approaches such as regression (Fu & Huang, 2008; Pan et al., 2018; Yang et al., 2018; Li et al., 2019), classification (Liu et al., 2020; Polat et al., 2022b; Vargas et al., 2022), and ranking-based methods (Niu et al., 2016; Chen et al., 2017; Cao et al., 2020; Shi et al., 2023) have been developed to capture this ordinal structure, often outperforming traditional frameworks by better aligning with the data's inherent order. However, insufficient attention to calibration has led to unreliable confidence estimates, particularly in fields like medical diagnosis, where the consequences of miscalibration can be severe (Guo et al., 2017).

Calibration aims to align a model's confidence estimates with actual accuracy, ensuring that predicted probabilities reflect the likelihood of correct predictions. Without proper calibration, models risk overconfidence, especially when encountering ambiguous or noisy data, which can lead to unsafe decisions in sensitive domains such as healthcare (Neumann et al., 2018; Moon et al., 2020). For example, in disease severity prediction, a model should adjust its confidence based on input uncertainty, particularly in ambiguous cases.

In addition to calibration, unimodality is crucial in ordinal classification. Unimodal distributions ensure that the model assigns the highest probability to the correct label, with probabilities gradually decreasing as the distance from the true label increases, preventing paradoxical or inconsistent predictions (Li et al., 2022; Vargas et al., 2022).

To address these dual challenges of calibration and unimodality, we propose Oridnal Regression loss for Calibration and Unimodality (ORCU), a novel loss function that integrates order-aware calibration with a unimodal regularization term. Traditional regularization techniques often neglect the ordered structure of ordinal tasks, but ORCU enforces both calibration and unimodality by explicitly modeling the ordinal relationships between classes. This ensures well-calibrated confidence estimates that reflect the full ordinal structure, enabling the model to produce unimodal probabil-

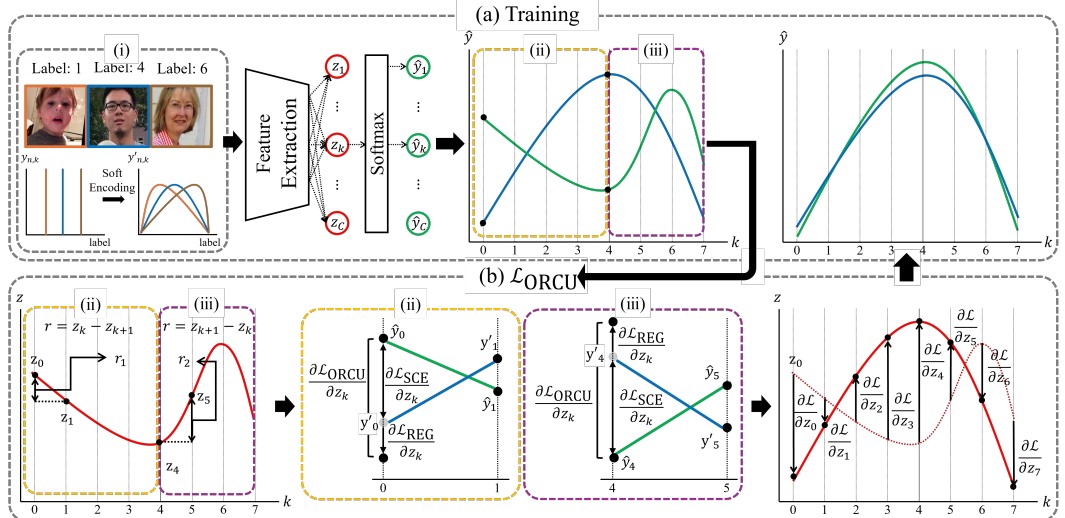

Figure 1: Overview of how the ORCU achieves improved calibration and unimodal probability distribution in ordinal classification. It illustrates the role of the calibration regularization term, $\mathcal{L}_{\text{REG}}$, when applied to soft label encoding for the Adience dataset (8 ordinal classes representing age groups) as an example. Soft encoding is applied to hard ordinal labels (a-i). Then, $\mathcal{L}_{\text{REG}}$ is applied differently for $k < y_n$ (a-ii) and $k \geq y_n$ (a-iii). (b) illustrates how $\mathcal{L}_{\text{ORCU}}$'s gradients are computed with examples shown for $k = 0, 4$. See section 3.4 and Table 1 for gradient analysis.

ity distributions that smoothly decrease as the distance from the true label increases. As a result, ORCU not only improves prediction accuracy but also enhances the reliability and trustworthiness of predictions, particularly in high-stakes applications like medical diagnosis.

**Contributions** We make the following key contributions: **(1)** We propose ORCU, a novel loss function that unifies calibration and unimodality within a single framework for ordinal regression. ORCU incorporates soft encoding and introduces a regularization term that uniquely applies order-aware conditions, ensuring that predicted probabilities reflect the ordinal relationships between classes while providing reliable confidence estimates. **(2)** We provide a comprehensive analysis demonstrating how ORCU effectively balances the trade-off between well-calibrated confidence estimates and accurate predictions. By integrating calibration and unimodality constraints into a single loss function, ORCU addresses the limitations of existing methods that focus solely on either calibration or ordinal structure. **(3)** By unifying calibration and unimodality constraints, we establish a new benchmark for reliable ordinal classification, significantly advancing the state of the art and paving the way for future research on trustworthy models for ordinal tasks.

## 2 RELATED WORKS

**Ordinal Regression** Ordinal regression addresses the challenge of predicting a target value with an inherent order, unlike nominal classification, where no ordinal relationships exist between labels. Given an input $x$, the goal is to predict a label $y$, which follows an ordinal relationship such that $y_1 \prec y_2 \prec \cdots \prec y_C$, with $\prec$ representing a label order (e.g., $y_i$ is less severe or earlier in rank than $y_{i+1}$). Approaches to ordinal classification are broadly divided into three categories: regression methods, classification-based methods, and ranking-based methods. In regression methods (Fu & Huang, 2008; Pan et al., 2018; Yang et al., 2018; Li et al., 2019), ordinal labels $y$ are treated as continuous variables, applying losses like L1 or L2 to predict a scalar value reflecting the label ordering. Classification-based methods (Liu et al., 2020; Polat et al., 2022b; Vargas et al., 2022) discretize the continuous target space into bins, treating each bin as a class $y \in \{1, 2, \ldots, C\}$ and predicting the class directly. Ranking-based methods (Niu et al., 2016; Chen et al., 2017; Cao et al., 2020; Shi et al., 2023) decompose the task into $C - 1$ binary classifiers, each determining whether the true label $y$ exceeds a threshold, capturing the ordinal relationships between labels.

**Loss functions for ordinal regression**    The inherent limitations of Cross-Entropy (CE) loss in handling ordinal relationships have led to several modified approaches (see Section 3.1). One such approach is Soft ORDinal (SORD) encoding method (Diaz & Marathe, 2019), which adjusts the label distribution by using soft labels to capture the proximity between classes, thereby ensuring smoother and more order-aware predictions (see Section 3.2). Class Distance Weighted Cross-Entropy (CDW-CE) (Polat et al., 2022b), on the other hand, keeps the traditional label structure but incorporates a distance-based penalty into the loss function, encouraging predictions closer to the true class and better aligned with the ordinal structure. CO2 (Albuquerque et al., 2021) extends CE with a regularization term enforcing unimodality, ensuring that predicted probabilities decrease smoothly as the distance from the true label increases. In contrast, Probabilistic Ordinal Embeddings (POE) (Li et al., 2021) introduces both a regularization term and architectural changes, representing each label as a probability distribution, thereby modeling both uncertainty and ordinal relationships.

**Regularization-based loss functions for calibration**    Calibration, which aims to align predicted confidence with actual accuracy, is particularly critical in high-risk tasks involving ordinal categories. Several regularization-based approaches have been proposed to improve calibration during training, as they preemptively address miscalibration without relying on post-hoc adjustments. Label Smoothing (LS) (Szegedy et al., 2016) is one of the foundational techniques, softening the sharp one-hot label distribution to mitigate overconfidence. Sample-dependent Focal Loss (FLSD) (Mukhoti et al., 2020) builds upon this by focusing calibration on harder-to-classify examples. Margin-based Label Smoothing (MbLS) (Liu et al., 2022) selectively smooths predictions based on the margin between predicted logits and true labels, while Multi-class Difference in Confidence and Accuracy (MDCA) (Hebbalaguppe et al., 2022) applies this margin adjustment across the entire predicted distribution. Adaptive and Conditional Label Smoothing (ACLS) (Park et al., 2023) dynamically adjusts the level of smoothing, applying stronger smoothing to miscalibrated predictions while preserving accurate confidence estimates for well-calibrated ones. While regularization-based calibration methods have been extensively studied, their application to ordinal tasks remains underexplored, despite the crucial role of confidence estimation in such settings.

## 3    A UNIFIED LOSS FUNCTIONS FOR CALIBRATION AND UNIMODALITY

### 3.1    LIMITATIONS OF CROSS-ENTROPY IN ORDINAL REGRESSION

In ordinal classification tasks, such as medical diagnosis or rating assessments, the inherent order between classes plays a crucial role. Traditional loss functions, like CE, often fail to capture this ordinal structure, leading to suboptimal performance and poorly calibrated predictions. This limitation is particularly problematic in high-risk applications where both accurate predictions and reliable confidence estimates are essential for informed decision-making.

Let $y_n \in \{1, \ldots, C\}$ denote the true class label of the $n$-th sample, where $N$ is the total number of samples and $C$ represents the number of classes. The CE loss is defined as $\mathcal{L}_{\text{CE}} = -\sum_{n=1}^{N} \sum_{k=1}^{C} y_{n,k} \log(\hat{y}_{n,k})$ where $y_{n,k}$ is the one-hot encoded true label and $\hat{y}_{n,k}$ is the predicted probability. The gradient of $\mathcal{L}_{\text{CE}}$ with respect to the logit $z_{n,k}$ is: $\frac{\partial \mathcal{L}_{\text{CE}}}{\partial z_{n,k}} = \hat{y}_{n,k} - y_{n,k}$. This formulation forces the model to focus sharply on the true label, often resulting in overconfident predictions that ignore relationships between adjacent classes. In ordinal tasks, predictions should reflect the ordered nature of classes, with probability mass distributed smoothly around the true label. However, $\mathcal{L}_{\text{CE}}$ typically produces sharp peaks, disregarding the ordinal structure and potentially leading to unreliable predictions in critical scenarios.

To address these limitations, we propose a novel loss function that explicitly accounts for ordinal relationships between classes while incorporating a regularization term to enforce unimodality and improve calibration. Our approach aims to balance accurate ordinal predictions with well-calibrated confidence estimates, enhancing performance in sensitive applications without compromising overall predictive accuracy.

### 3.2    SOFT ORDINAL ENCODING

One of the primary limitations of the traditional $\mathcal{L}_{\text{CE}}$ loss in ordinal tasks is its reliance on one-hot encoding, which concentrates probability mass entirely on the true label. This results in overconfi-

dent predictions, disregarding the relationships between adjacent classes. To address these issues, we employ the SORD encoding method (Diaz & Marathe, 2019), which redistributes the probability mass across the ordinal classes in a way that reflects both the inherent ordinal structure and the uncertainty between adjacent labels.

In an ordinal classification problem with $C$ classes, the true label $y_n$ for the $n$-th sample is represented as a soft-encoded probability distribution $y'_{n,k}$, defined as $y'_{n,k} = \frac{e^{-\phi(y_n, r_k)}}{\sum_{j=1}^{C} e^{-\phi(y_n, r_j)}}$, for $k = 1, \ldots, C$, where $\phi(y_n, r_k)$ is a distance metric that penalizes deviations from the true class $y_n$ to each ordinal class $r_k$ during the soft encoding process. The Soft-encoded Cross-Entropy (SCE) loss is then defined as:

$$\mathcal{L}_{\text{SCE}} = -\sum_{n=1}^{N} \sum_{k=1}^{C} y'_{n,k} \log(\hat{y}_{n,k}), \tag{1}$$

The gradient of $\mathcal{L}_{\text{SCE}}$ with respect to the logit $z_{n,k}$ is $\frac{\partial \mathcal{L}_{\text{SCE}}}{\partial z_{n,k}} = \hat{y}_{n,k} - y'_{n,k}$. This gradient encourages the model to distribute probability across adjacent classes in a way that aligns with the soft-encoded true label distribution, mitigating overconfidence and resulting in more calibrated predictions that accurately reflect the ordinal relationships between classes.

## 3.3 ORDER-AWARE REGULARIZATION: ENFORCING UNIMODALITY AND CALIBRATION

While $\mathcal{L}_{\text{SCE}}$ effectively captures the ordinal relationships between labels and mitigates the overconfidence issue in $\mathcal{L}_{\text{CE}}$, it can still lead to underconfident predictions due to the redistribution of probability mass across classes. To address this issue, we introduce a regularization-based method that simultaneously enhances the model's calibration and reinforces the learning of the ordinal structure (which inherently leads to unimodality in the output distribution).

Unlike traditional regularization methods, which focus solely on the highest probability class and ignore the relationships between adjacent labels (Park et al., 2023), our approach leverages the ordinal structure of the task. Specifically, our method adjusts the logits based on their ordinal relationship to the true label $y_n$, ensuring that the calibration regularization is order-aware. This approach divides the logits into two regions: those where the class index $k$ is smaller than the true label and those where $k$ is greater than or equal to the true label. By doing so, the regularization ensures smooth decreases in probability on both sides of the true label, enforcing a unimodal distribution while simultaneously improving calibration. The specific formulation of the regularization term is as follows:

$$\mathcal{L}_{\text{REG}} = \sum_{n=1}^{N} \sum_{k=1}^{C-1} \begin{cases} \hat{I}(z_{n,k} - z_{n,k+1}), & \text{if } k < y_n, \\ \hat{I}(z_{n,k+1} - z_{n,k}), & \text{if } k \geq y_n, \end{cases} \tag{2}$$

where $\hat{I}(r) = -\frac{1}{t}\log(-r)$ if $r \leq -\frac{1}{t^2}$, and $\hat{I}(r) = tr - \frac{1}{t}\log\left(\frac{1}{t^2}\right) + \frac{1}{t}$ otherwise.

This penalty function applies strong corrections when the differences between adjacent logits approach the boundary value $-1/t^2$, ensuring unimodality is preserved. The temperature parameter $t$ controls the strength of this regularization. The final loss function ORCU, is defined as:

$$\mathcal{L}_{\text{ORCU}} = \mathcal{L}_{\text{SCE}} + \mathcal{L}_{\text{REG}}. \tag{3}$$

By integrating both $\mathcal{L}_{\text{SCE}}$ and $\mathcal{L}_{\text{REG}}$, $\mathcal{L}_{\text{ORCU}}$ ensures that the model produces well-calibrated, unimodal predictions that accurately reflect the ordinal relationships between classes.

## 3.4 GRADIENT ANALYSIS: ENFORCING UNIMODALITY AND CALIBRATION

The gradient behavior of the proposed regularization term $\mathcal{L}_{\text{REG}}$ is crucial for ensuring both unimodality and calibration. As illustrated in Figure 1, the $\mathcal{L}_{\text{ORCU}}$ loss applies soft label encoding and uses regularization differently depending on whether $k < y_n$ or $k \geq y_n$. To better understand its impact, we divide the gradient into four cases based on the relationship between the class index $k$ and the target label $y_n$, as well as the magnitude of the difference between adjacent logits $r$ (Table 1-b). The regularization term adjusts the logits to ensure that the output distribution remains smooth and unimodal, particularly in ordinal classification tasks.

When $k < y_n$ (Figure 1-ii), the model adjusts the logits to ensure that the probability distribution decreases smoothly as the distance from the true label increases, maintaining a unimodal structure.

Table 1: Gradient analysis of our loss function. The gradient is computed w.r.t. a logit when $k < y_n$ and $k \geq y_n$. We also show the gradient of each term, $\mathcal{L}_{\text{SCE}}$ and $\mathcal{L}_{\text{REG}}$, to analyze each one's contribution. $\hat{y}_{n,k}$ denote the predicted probability, $y'_{n,k}$ representing the soft-encoded target, and $t$ being a variable controlling the regularization strength. Note that $\frac{\partial \mathcal{L}_{\text{ORCU}}}{\partial z_k} = \frac{\partial \mathcal{L}_{\text{SCE}}}{\partial z_k} + \frac{\partial \mathcal{L}_{\text{REG}}}{\partial z_k}$.

| | | $k < y_n$ | | $k \geq y_n$ | |
| | | $(r = z_{n,k} - z_{n,k+1})$ | | $(r = z_{n,k+1} - z_{n,k})$ | |
| | | $r \leq -\frac{1}{t^2}$ | $r > -\frac{1}{t^2}$ | $r \leq -\frac{1}{t^2}$ | $r > -\frac{1}{t^2}$ |
|---|---|---|---|---|---|
| (a) | $\frac{\partial \mathcal{L}_{\text{SCE}}}{\partial z_{n,k}}$ | $\hat{y}_{n,k} - y'_{n,k}$ | | | |
| (b) | $\frac{\partial \mathcal{L}_{\text{REG}}}{\partial z_{n,k}}$ | $-\frac{1}{tr}$ | $t$ | $\frac{1}{tr}$ | $-t$ |
| (c) | $\frac{\partial \mathcal{L}_{\text{ORCU}}}{\partial z_{n,k}}$ | $\hat{y}_{n,k} - (y'_{n,k} + \frac{1}{tr})$ | $\hat{y}_{n,k} - (y'_{n,k} - t)$ | $\hat{y}_{n,k} - (y'_{n,k} - \frac{1}{tr})$ | $\hat{y}_{n,k} - (y'_{n,k} + t)$ |

This is achieved by keeping the difference between adjacent logits, $r$, negative. If $r$ is negative and has a large absolute value, the gradient of $\mathcal{L}_{\text{REG}}$ becomes smaller, indicating that the model is close to achieving a unimodal distribution. However, as $r$ approaches the boundary $-\frac{1}{t^2}$, the gradient increases sharply to prevent any violations of unimodality. On the other hand, when $r > -\frac{1}{t^2}$, indicating a deviation from the desired unimodal structure, a constant penalty is applied to restore the distribution.

**Dynamic gradient adjustment for simultaneous calibration and unimodality**  When the gradients of $\mathcal{L}_{\text{SCE}}$ and $\mathcal{L}_{\text{REG}}$ are combined, the regularization term enables the model to address both calibration and the learning of the ordinal structure (i.e., unimodality) simultaneously.

First, consider the case where $k < y_n$ (as shown in Figure 1-(a-ii)), where the predicted class is lower than the true class. In this scenario, for large positive $r$ values (Figure 1-(b-ii)), the regularization term reduces $y'_{n,k}$ by a factor of $t$, increasing the difference between $\hat{y}_{n,k}$ and $y'_{n,k} - t$. This larger gradient leads to a more substantial update to the logit $z_k$, compared to using $\mathcal{L}_{\text{SCE}}$ alone. Consequently, this adjustment not only restores unimodality by driving $r$ towards negative values but also helps correct overconfident predictions for incorrect labels. By increasing the gradient for such incorrect predictions, the model is able to reduce the predicted probability for the incorrect class more effectively, improving overall calibration and ordinal structure learning.

Next, consider the case where $k = y_n$, and $r > -\frac{1}{t^2}$ (as shown in Figure 1-(b-iii)). In this underconfident scenario, where the predicted probability $\hat{y}_{n,k}$ is lower than the target distribution $y'_{n,k}$, the regularization term modifies the gradient to $\hat{y}_{n,k} - (y'_{n,k} + t)$. This larger gradient (with a greater absolute value) results in a more substantial update to the logit $z_k$, increasing the predicted probability for the true class $y_n$. At the same time, this adjustment restores unimodality by ensuring that $r$ remains negative, aligning the probability distribution smoothly around the true label.

By combining these two examples, we demonstrate that the proposed $\mathcal{L}_{\text{REG}}$ term dynamically adjusts the gradients based on the value of $r$, enabling the model to simultaneously achieve well-calibrated predictions and maintain the ordinal structure (unimodality) of the task. Whether the model is overconfident or underconfident, the regularization term ensures appropriate updates to the logits, balancing both calibration and the preservation of the ordinal relationships between classes. This dual mechanism is essential for high-risk ordinal tasks, where both accurate predictions and reliable confidence estimates are critical.

## 4 EXPERIMENTS

### 4.1 DATASETS AND IMPLEMENTATION DETAILS

We evaluated the proposed $\mathcal{L}_{\text{ORCU}}$ loss function on three public datasets selected for their ordinal nature and sufficient sample sizes, covering diverse tasks such as age estimation, image aesthetics assessment, and medical diagnosis. The Adience dataset (Eidinger et al., 2014), containing 26,580 images categorized into 8 age groups, was used for age estimation, employing five-fold cross-validation

with an 80/20 train-test split[1]. For image aesthetics assessment, the Image Aesthetics dataset (Schifanella et al., 2015), consisting of 13,364 images with five ordinal labels, was used with five-fold cross-validation and an 80/20 train-test split. The LIMUC dataset (Polat et al., 2022a), comprising 11,276 images from ulcerative colitis patients with four mayo endoscopic scores (MES), was used for medical diagnosis, employing an 85/15 subject-exclusive (Paplhám et al., 2024) train-test split and a ten-fold cross-validation protocol (Polat et al., 2022b).

For all tasks, we used a ResNet-50 architecture (He et al., 2015) pretrained on ImageNet (Russakovsky et al., 2015). We applied a layer-wise learning rate strategy (0.01 for the fully connected layer, 0.001 for others), using AdamW (Loshchilov & Hutter, 2019) for optimization. Training was conducted for 100 epochs with a batch size of 64, applying image augmentations via Albumentations (Buslaev et al., 2020). The squared distance metric was used for soft encoding, with the temperature parameter $t$ initialized at 10.0 and gradually decreased throughout training. All experiments were conducted on NVIDIA RTX 4090 GPUs.

## 4.2 Performance Metrics

Evaluating calibration is crucial in ordinal classification tasks, where models must provide not only accurate predictions but also reliable confidence estimates. We focus on two specific calibration metrics: static calibration error (SCE) and adaptive calibration error (ACE) [2], which are particularly suited to addressing class imbalances and capturing calibration across all predictions. Although expected calibration error (ECE) is commonly used, it can be misleading, especially in imbalanced datasets where confidence is skewed towards high-probability predictions, failing to accurately reflect model performance in all classes (Nixon et al., 2019). SCE and ACE provide more robust measures of calibration by evaluating calibration per class or dynamically adjusting bin sizes.

To capture ordinal relationships between classes, we use quadratic weighted kappa (QWK) as the primary metric, as it penalizes larger misclassifications more heavily and better reflects the distance between predicted and true labels compared to accuracy. Although accuracy is reported, QWK serves as the primary metric for assessing classification performance in ordinal tasks.

## 4.3 Results and Analysis

Our proposed loss function $\mathcal{L}_{\text{ORCU}}$ demonstrates superior performance across both calibration and ordinal regression metrics, surpassing baseline methods designed for either objective individually (Table 2). Unlike existing loss functions that primarily focus on a single goal—either calibration or classification—$\mathcal{L}_{\text{ORCU}}$ effectively balances both, achieving significant improvements in calibration metrics while maintaining strong predictive accuracy.

In high-stakes tasks such as medical diagnosis and severity grading, where accurate and well-calibrated predictions are critical, $\mathcal{L}_{\text{ORCU}}$ proves particularly advantageous. Existing ordinal loss functions, such as CE and its variants, tend to overlook calibration error, while traditional calibration losses fail to account for the ordinal structure that is intrinsic to these tasks. To thoroughly evaluate $\mathcal{L}_{\text{ORCU}}$, we compared it against 10 baseline loss functions, categorized into two groups: ordinal loss functions—CE, SORD (Diaz & Marathe, 2019), CDW-CE (Polat et al., 2022b), CO2 (Albuquerque et al., 2021), and POE (Li et al., 2021)—and calibration loss functions—LS (Szegedy et al., 2016), FLSD (Mukhoti et al., 2020), MbLS (Liu et al., 2022), MDCA (Hebbalaguppe et al., 2022), and ACLS (Park et al., 2023). Each loss function was evaluated under identical experimental conditions to ensure fair benchmarking.

### 4.3.1 Comparison with Loss Functions Targeting Individual Objectives

**Comparison with Ordinal Loss Functions**    Our method, $\mathcal{L}_{\text{ORCU}}$, consistently outperforms all baseline ordinal loss functions, providing superior calibration and classification performance. As detailed in Table 2, $\mathcal{L}_{\text{ORCU}}$ achieves the lowest SCE, ACE, and ECE scores, while maintaining competitive accuracy and QWK scores. This highlights its ability to deliver well-calibrated predictions without compromising on predictive performance. Traditional ordinal loss function, primarily focus

---

[1] https://github.com/GilLevi/AgeGenderDeepLearning
[2] https://github.com/Jonathan-Pearce/calibration_library

Table 2: Calibration and accuracy performance for different loss functions on three popular ordinal regression benchmarks. Competing methods in two areas, i.e., ordinal regression and network calibration, are compared with ours. We use ResNet-50 for classifications and use 15 bins for calibration metrics calculation. The measures are presented as the mean and standard deviation over all folds.

| Evaluation Metrics / Loss | SCE($\downarrow$) | ACE($\downarrow$) | ECE($\downarrow$) | Acc($\uparrow$) | QWK($\uparrow$) |
|---|---|---|---|---|---|
| Adience ($n =$17,423) | | | | | |
| Cross Entropy | $0.8495 \pm 0.0033$ | $0.8356 \pm 0.0068$ | $0.3364 \pm 0.0401$ | $0.5639 \pm 0.0486$ | $0.8795 \pm 0.0345$ |
| SORD (Diaz & Marathe, 2019) | $0.7823 \pm 0.0105$ | $0.7783 \pm 0.0102$ | $0.0731 \pm 0.0240$ | $\mathbf{0.5910 \pm 0.0439}$ | $0.8995 \pm 0.0291$ |
| CDW-CE (Polat et al., 2022b) | $0.8429 \pm 0.0062$ | $0.8372 \pm 0.0071$ | $0.2913 \pm 0.0210$ | $0.5789 \pm 0.0362$ | $0.8988 \pm 0.0257$ |
| CO2 (Albuquerque et al., 2021) | $0.8521 \pm 0.0055$ | $0.8368 \pm 0.0080$ | $0.3533 \pm 0.0406$ | $0.5637 \pm 0.0489$ | $0.8728 \pm 0.0418$ |
| POE (Li et al., 2021) | $0.8340 \pm 0.0042$ | $0.8244 \pm 0.0065$ | $0.2733 \pm 0.0384$ | $0.5652 \pm 0.0522$ | $0.8669 \pm 0.0395$ |
| LS (Szegedy et al., 2016) | $0.8195 \pm 0.0071$ | $0.8101 \pm 0.0072$ | $0.1890 \pm 0.0415$ | $0.5792 \pm 0.0495$ | $0.8824 \pm 0.0334$ |
| FLSD (Mukhoti et al., 2020) | $0.8474 \pm 0.0063$ | $0.8340 \pm 0.0062$ | $0.3193 \pm 0.0407$ | $0.5718 \pm 0.0532$ | $0.8840 \pm 0.0347$ |
| MbLS (Liu et al., 2022) | $0.8400 \pm 0.0043$ | $0.8323 \pm 0.0052$ | $0.2815 \pm 0.0344$ | $0.5765 \pm 0.0489$ | $0.8894 \pm 0.0324$ |
| MDCA(Hebbalaguppe et al., 2022) | $0.8561 \pm 0.0527$ | $0.8365 \pm 0.0059$ | $0.3372 \pm 0.0411$ | $0.5676 \pm 0.0488$ | $0.8770 \pm 0.0346$ |
| ACLS (Park et al., 2023) | $0.8398 \pm 0.0040$ | $0.8295 \pm 0.0045$ | $0.2847 \pm 0.0378$ | $0.5762 \pm 0.0488$ | $0.8788 \pm 0.0364$ |
| ORCU (Ours) | $\mathbf{0.4598 \pm 0.0435}$ | $\mathbf{0.4565 \pm 0.0437}$ | $0.0583 \pm 0.0279$ | $0.5878 \pm 0.0426$ | $\mathbf{0.9036 \pm 0.0281}$ |
| Image Aesthetics ($n =$13,364) | | | | | |
| Cross Entropy | $0.7637 \pm 0.0037$ | $0.7558 \pm 0.0035$ | $0.2057 \pm 0.0162$ | $0.7030 \pm 0.0080$ | $0.4961 \pm 0.0199$ |
| SORD (Diaz & Marathe, 2019) | $0.6844 \pm 0.0018$ | $0.6833 \pm 0.0018$ | $0.1846 \pm 0.0026$ | $0.7092 \pm 0.0044$ | $0.5030 \pm 0.0030$ |
| CDW-CE (Polat et al., 2022b) | $0.7519 \pm 0.0030$ | $0.7480 \pm 0.0026$ | $0.1751 \pm 0.0136$ | $0.7041 \pm 0.0064$ | $0.4837 \pm 0.0148$ |
| CO2 (Albuquerque et al., 2021) | $0.7699 \pm 0.0032$ | $0.7614 \pm 0.0036$ | $0.2269 \pm 0.0151$ | $0.6980 \pm 0.0072$ | $0.4919 \pm 0.0158$ |
| POE (Li et al., 2021) | $0.7559 \pm 0.0020$ | $0.7493 \pm 0.0035$ | $0.1883 \pm 0.0111$ | $0.7010 \pm 0.0097$ | $0.4909 \pm 0.0156$ |
| LS (Szegedy et al., 2016) | $0.7222 \pm 0.0010$ | $0.7179 \pm 0.0014$ | $\mathbf{0.0991 \pm 0.0062}$ | $0.7063 \pm 0.0054$ | $0.4990 \pm 0.0202$ |
| FLSD (Mukhoti et al., 2020) | $0.7519 \pm 0.0030$ | $0.7482 \pm 0.0026$ | $0.1751 \pm 0.0136$ | $0.7013 \pm 0.0075$ | $0.4961 \pm 0.0147$ |
| MbLS (Liu et al., 2022) | $0.7577 \pm 0.0016$ | $0.7510 \pm 0.0020$ | $0.1895 \pm 0.0062$ | $0.7040 \pm 0.0057$ | $0.4942 \pm 0.0159$ |
| MDCA(Hebbalaguppe et al., 2022) | $0.7620 \pm 0.0033$ | $0.7549 \pm 0.0033$ | $0.2022 \pm 0.0124$ | $0.7031 \pm 0.0110$ | $0.4993 \pm 0.0129$ |
| ACLS (Park et al., 2023) | $0.7595 \pm 0.0010$ | $0.7535 \pm 0.0013$ | $0.1957 \pm 0.0097$ | $0.7030 \pm 0.0091$ | $0.4953 \pm 0.0147$ |
| ORCU (Ours) | $\mathbf{0.6805 \pm 0.0045}$ | $\mathbf{0.6794 \pm 0.0046}$ | $0.1082 \pm 0.0312$ | $\mathbf{0.7113 \pm 0.0038}$ | $\mathbf{0.5188 \pm 0.0139}$ |
| LIMUC ($n =$11,276) | | | | | |
| Cross Entropy | $0.6997 \pm 0.0076$ | $0.6948 \pm 0.0075$ | $0.1295 \pm 0.0170$ | $0.7702 \pm 0.0066$ | $0.8461 \pm 0.0090$ |
| SORD (Diaz & Marathe, 2019) | $0.6382 \pm 0.0031$ | $0.6370 \pm 0.0032$ | $0.1636 \pm 0.0064$ | $0.7749 \pm 0.0060$ | $0.8539 \pm 0.0062$ |
| CDW-CE (Polat et al., 2022b) | $0.6980 \pm 0.0048$ | $0.6927 \pm 0.0042$ | $0.1190 \pm 0.0096$ | $0.7773 \pm 0.0058$ | $0.8551 \pm 0.0069$ |
| CO2 (Albuquerque et al., 2021) | $0.7105 \pm 0.0044$ | $0.7042 \pm 0.0042$ | $0.1544 \pm 0.0118$ | $0.7662 \pm 0.0076$ | $0.8411 \pm 0.0081$ |
| POE (Li et al., 2021) | $0.6933 \pm 0.0043$ | $0.6881 \pm 0.0046$ | $0.1149 \pm 0.0105$ | $0.7724 \pm 0.0040$ | $0.8353 \pm 0.0110$ |
| LS (Szegedy et al., 2016) | $0.6647 \pm 0.0010$ | $0.6603 \pm 0.0022$ | $\mathbf{0.0592 \pm 0.0088}$ | $0.7633 \pm 0.0053$ | $0.8402 \pm 0.0096$ |
| FLSD (Mukhoti et al., 2020) | $0.6674 \pm 0.0016$ | $0.6631 \pm 0.0022$ | $0.1069 \pm 0.0167$ | $0.7657 \pm 0.0059$ | $0.8459 \pm 0.0110$ |
| MbLS (Liu et al., 2022) | $0.6988 \pm 0.0022$ | $0.6982 \pm 0.0035$ | $0.1301 \pm 0.0095$ | $0.7665 \pm 0.0057$ | $0.8490 \pm 0.0113$ |
| MDCA(Hebbalaguppe et al., 2022) | $0.6934 \pm 0.0074$ | $0.6879 \pm 0.0061$ | $0.1197 \pm 0.0129$ | $0.7683 \pm 0.0044$ | $0.8443 \pm 0.0118$ |
| ACLS (Park et al., 2023) | $0.6995 \pm 0.0030$ | $0.6939 \pm 0.0032$ | $0.1299 \pm 0.0140$ | $0.7683 \pm 0.0064$ | $0.8454 \pm 0.0092$ |
| ORCU (Ours) | $\mathbf{0.5205 \pm 0.0098}$ | $\mathbf{0.5182 \pm 0.0107}$ | $0.0853 \pm 0.0269$ | $\mathbf{0.7785 \pm 0.0064}$ | $\mathbf{0.8578 \pm 0.0048}$ |

on accuracy but tend to produce overconfident predictions by concentrating probability mass on the true label (Figure 2-a, c-e). Although SORD reduces overconfidence by distributing probability mass across adjacent labels, it results in underconfident predictions (Figure 2-c), limiting its effectiveness. In contrast, $\mathcal{L}_{\text{ORCU}}$ effectively balances these extremes. As shown in Figure 2-f, our method mitigates both overconfidence and underconfidence, delivering well-calibrated confidence estimates. Additionally, high QWK scores across all datasets confirm that $\mathcal{L}_{\text{ORCU}}$ captures the inherent ordinal relationships between classes more effectively than baseline ordinal loss functions.

**Comparison with Calibration Loss Functions**    When compared with calibration-focused loss functions, $\mathcal{L}_{\text{ORCU}}$ also demonstrates superior performance. Calibration losses like LS, FLSD, MbLS, MDCA, and ACLS are primarily designed to reduce calibration error in nominal tasks, without considering the ordinal relationships between classes. However, $\mathcal{L}_{\text{ORCU}}$ incorporates these ordinal relationships into the calibration process, which leads to superior results in tasks that require both accurate predictions and well-calibrated confidence estimates.As seen in Table 2, $\mathcal{L}_{\text{ORCU}}$ consistently achieves the lowest SCE and ACE scores, while also delivering strong accuracy and QWK scores. Although LS slightly outperforms $\mathcal{L}_{\text{ORCU}}$ in ECE, it fails to account for the ordinal struc-

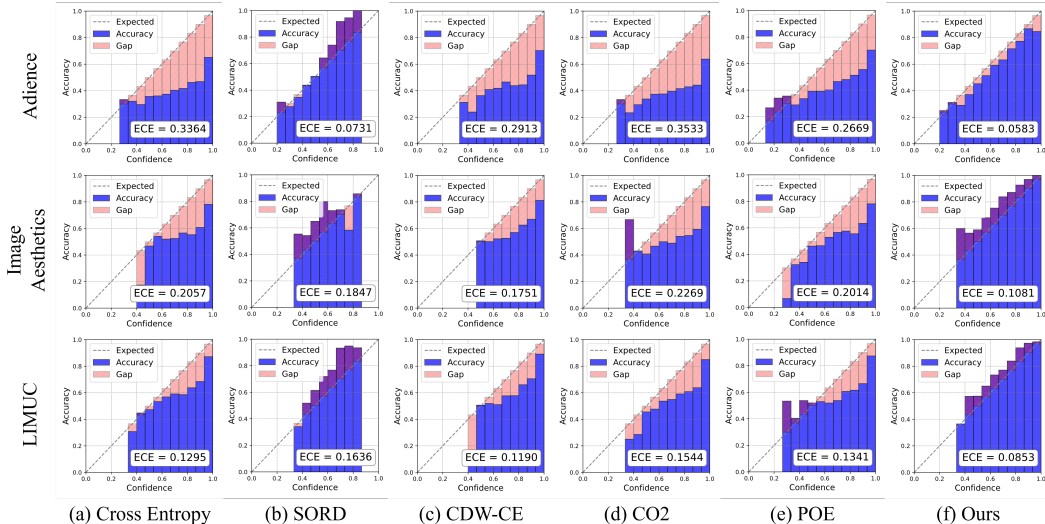

Figure 2: Reliability diagrams for different ordinal loss functions. The diagrams show model confidence alongside the calibration gap between confidence and accuracy, using the test split of Adience, Image Aesthetics, and LIMUC. Predictions above the diagonal, expected line indicate underconfidence, while those below the line represent overconfidence. ECE is computed using 15 bins.

ture, as evidenced by its lower QWK scores. This demonstrates the advantage of $\mathcal{L}_{\text{ORCU}}$ in ordinal classification tasks, where preserving the relationships between labels is critical.

In summary, $\mathcal{L}_{\text{ORCU}}$ provides a balanced solution for both calibration and classification, outperforming loss functions designed for either objective individually. By delivering well-calibrated predictions while maintaining high accuracy and QWK scores, $\mathcal{L}_{\text{ORCU}}$ effectively addresses the dual challenges of calibration and ordinal classification. This makes it particularly valuable in such critical tasks where both aspects are essential.

Table 3: Ablation study on different distance metrics used in the soft encoding. The table analyzes the performance impact of applying various distance metrics (absolute, Huber, exponential, and squared) across multiple datasets. The results are presented as the mean and standard deviation over all folds.

| Evaluation Metrics / Distance Metrics $\phi$ | SCE($\downarrow$) | ACE($\downarrow$) | ECE($\downarrow$) | Acc($\uparrow$) | QWK($\uparrow$) |
|---|---|---|---|---|---|
| Adience ($n =$17,423) | | | | | |
| Absolute | $0.7121 \pm 0.0088$ | $0.7167 \pm 0.0088$ | $0.0684 \pm 0.0203$ | $0.5800 \pm 0.0334$ | $0.8961 \pm 0.0299$ |
| Huber | $0.7274 \pm 0.0110$ | $0.7233 \pm 0.0115$ | $\mathbf{0.0553 \pm 0.0084}$ | $0.5704 \pm 0.0461$ | $0.8948 \pm 0.0323$ |
| Exponential | $0.7774 \pm 0.0120$ | $0.7753 \pm 0.0116$ | $0.0868 \pm 0.0262$ | $0.5794 \pm 0.0425$ | $0.8983 \pm 0.0320$ |
| Squared | $\mathbf{0.4598 \pm 0.0435}$ | $\mathbf{0.4565 \pm 0.0437}$ | $0.0583 \pm 0.0279$ | $\mathbf{0.5878 \pm 0.0426}$ | $\mathbf{0.9036 \pm 0.0281}$ |
| Image Aesthetics ($n =$13,364) | | | | | |
| Absolute | $0.6849 \pm 0.0020$ | $0.6837 \pm 0.0018$ | $0.1351 \pm 0.0153$ | $0.7101 \pm 0.0044$ | $0.5175 \pm 0.0117$ |
| Huber | $0.6827 \pm 0.0016$ | $0.6839 \pm 0.0034$ | $0.1460 \pm 0.0180$ | $0.7077 \pm 0.0083$ | $0.5127 \pm 0.0167$ |
| Exponential | $0.6869 \pm 0.0016$ | $0.6859 \pm 0.0014$ | $0.1798 \pm 0.0052$ | $\mathbf{0.7156 \pm 0.0036}$ | $\mathbf{0.5265 \pm 0.0125}$ |
| Squared | $\mathbf{0.6805 \pm 0.0045}$ | $\mathbf{0.6794 \pm 0.0046}$ | $\mathbf{0.1082 \pm 0.0312}$ | $0.7113 \pm 0.0038$ | $0.5188 \pm 0.0139$ |
| LIMUC ($n =$11,276) | | | | | |
| Absolute | $0.6414 \pm 0.0010$ | $0.6406 \pm 0.0011$ | $0.1552 \pm 0.0113$ | $\mathbf{0.7824 \pm 0.0021}$ | $\mathbf{0.8625 \pm 0.0033}$ |
| Huber | $0.6405 \pm 0.0037$ | $0.6399 \pm 0.0038$ | $0.1914 \pm 0.0136$ | $0.7809 \pm 0.0075$ | $0.8599 \pm 0.0059$ |
| Exponential | $0.6397 \pm 0.0036$ | $0.6391 \pm 0.0036$ | $0.2286 \pm 0.0150$ | $0.7794 \pm 0.0072$ | $0.8586 \pm 0.0055$ |
| Squared | $\mathbf{0.5205 \pm 0.0098}$ | $\mathbf{0.5182 \pm 0.0107}$ | $\mathbf{0.0853 \pm 0.0269}$ | $0.7785 \pm 0.0064$ | $0.8578 \pm 0.0048$ |

### 4.3.2 ABLATION STUDIES

**Impact of distance metrics on calibration and prediction** The choice of distance metric in soft encoding is a key factor affecting both calibration and prediction performance. Given our focus on

Table 4: Ablation study showing impacts of different calibration regularization methods in soft label encoding. Our method is compared with $\mathcal{L}_{\text{MbLS}}$, $\mathcal{L}_{\text{MDCA}}$, and $\mathcal{L}_{\text{ACLS}}$. The results demonstrate the importance of incorporating order-aware conditions in the regularization term, which leads to improved calibration.

| Combination / Evaluation Metrics | SCE($\downarrow$) | ACE($\downarrow$) | ECE($\downarrow$) | Acc($\uparrow$) | QWK($\uparrow$) |
|---|---|---|---|---|---|
| Adience ($n =$17,423) | | | | | |
| $\mathcal{L}_{\text{SCE}} + \mathcal{L}_{\text{MbLS\_REG}}$ | $0.7823 \pm 0.0092$ | $0.7788 \pm 0.0092$ | $0.0711 \pm 0.0217$ | $0.5906 \pm 0.0418$ | $0.8988 \pm 0.0298$ |
| $\mathcal{L}_{\text{SCE}} + \mathcal{L}_{\text{MDCA\_REG}}$ | $0.7849 \pm 0.0091$ | $0.7810 \pm 0.0100$ | $0.0677 \pm 0.0315$ | $\mathbf{0.5975 \pm 0.0470}$ | $0.9024 \pm 0.0289$ |
| $\mathcal{L}_{\text{SCE}} + \mathcal{L}_{\text{ACLS\_REG}}$ | $0.7827 \pm 0.0099$ | $0.7791 \pm 0.0100$ | $0.0746 \pm 0.0239$ | $0.5936 \pm 0.0453$ | $0.9000 \pm 0.0322$ |
| $\mathcal{L}_{\text{SCE}} + \mathcal{L}_{\text{ORCU\_REG}}$ (Ours) | $\mathbf{0.4598 \pm 0.0435}$ | $\mathbf{0.4565 \pm 0.0437}$ | $\mathbf{0.0583 \pm 0.0279}$ | $0.5878 \pm 0.0426$ | $\mathbf{0.9036 \pm 0.0281}$ |
| Image Aesthetics ($n =$13,364) | | | | | |
| $\mathcal{L}_{\text{SCE}} + \mathcal{L}_{\text{MbLS\_REG}}$ | $0.6866 \pm 0.0021$ | $0.6852 \pm 0.0023$ | $0.1921 \pm 0.0057$ | $0.7140 \pm 0.0057$ | $0.5106 \pm 0.0116$ |
| $\mathcal{L}_{\text{SCE}} + \mathcal{L}_{\text{MDCA\_REG}}$ | $0.6867 \pm 0.0033$ | $0.6859 \pm 0.0035$ | $0.1501 \pm 0.0080$ | $\mathbf{0.7156 \pm 0.0088}$ | $0.4960 \pm 0.0192$ |
| $\mathcal{L}_{\text{SCE}} + \mathcal{L}_{\text{ACLS\_REG}}$ | $0.6836 \pm 0.0028$ | $0.6819 \pm 0.0033$ | $0.1829 \pm 0.0081$ | $0.7058 \pm 0.0082$ | $0.5033 \pm 0.0130$ |
| $\mathcal{L}_{\text{SCE}} + \mathcal{L}_{\text{ORCU\_REG}}$ (Ours) | $\mathbf{0.6805 \pm 0.0045}$ | $\mathbf{0.6794 \pm 0.0046}$ | $\mathbf{0.1082 \pm 0.0312}$ | $0.7113 \pm 0.0038$ | $\mathbf{0.5188 \pm 0.0139}$ |
| LIMUC ($n =$11,276) | | | | | |
| $\mathcal{L}_{\text{SCE}} + \mathcal{L}_{\text{MbLS\_REG}}$ | $0.6398 \pm 0.0026$ | $0.6386 \pm 0.0026$ | $0.1662 \pm 0.0056$ | $0.7784 \pm 0.0051$ | $0.8564 \pm 0.0035$ |
| $\mathcal{L}_{\text{SCE}} + \mathcal{L}_{\text{MDCA\_REG}}$ | $0.6387 \pm 0.0026$ | $0.6368 \pm 0.0027$ | $0.1373 \pm 0.0065$ | $0.7725 \pm 0.0056$ | $0.8544 \pm 0.0043$ |
| $\mathcal{L}_{\text{SCE}} + \mathcal{L}_{\text{ACLS\_REG}}$ | $0.6366 \pm 0.0029$ | $0.6349 \pm 0.0030$ | $0.1590 \pm 0.0067$ | $0.7710 \pm 0.0061$ | $0.8532 \pm 0.0052$ |
| $\mathcal{L}_{\text{SCE}} + \mathcal{L}_{\text{ORCU\_REG}}$ (Ours) | $\mathbf{0.5205 \pm 0.0098}$ | $\mathbf{0.5182 \pm 0.0107}$ | $\mathbf{0.0853 \pm 0.0269}$ | $\mathbf{0.7785 \pm 0.0064}$ | $\mathbf{0.8578 \pm 0.0048}$ |

achieving reliable, well-calibrated predictions in high-risks ordinal regression tasks, we compared four common distance metrics: Absolute, Squared, Huber, and Exponential. As shown in Table 3, the Squared distance metric consistently provided the best calibration results, particularly in terms of ECE, ACE, and SCE. While Absolute and Exponential metrics showed competitive classification performance on some datasets, they delivered less reliable calibration, making them less suitable for high-risk tasks. In contrast, the Squared metric offered a balanced improvement in both calibration and prediction accuracy, making it the most effective choice for our loss function.

**Importance of order-aware regularization method**    We conducted an ablation study to assess the effect of different calibration regularization methods applied within the $\mathcal{L}_{\text{SCE}}$. Our approach, which integrates both calibration and unimodality constraints, demonstrated consistently superior calibration results, lower SCE, ACE, and ECE scores (see Table 4). Unlike traditional methods that focus calibration only on the highest probability class and overlook ordinal relationships, our method applies calibration across all labels, ensuring the entire ordinal structure is respected. This approach led to improved calibration metrics (SCE, ACE, ECE) and higher QWK scores, which better capture the model's ability to reflect the inherent order of labels. Although there was a slight decrease in accuracy, this minimal trade-off is offset by the significant gains in calibration and QWK, making it particularly valuable for high-risk, ordinal classification tasks.

## 5    CONCLUSION

We introduced ORCU, a unified loss function that integrates calibration and unimodality for ordinal regression. Calibration has been overlooked in the literature on ordinal tasks, despite its importance in high-risk applications. ORCU addresses the issue by explicitly targeting calibration improvement in ordinal regression, using comprehensive metrics such as SCE, ACE and ECE to evaluate its effectiveness. By leveraging soft ordinal encoding and order-aware regularization, which simultaneously enforces calibration and unimodality, ORCU balances accurate predictions with well-calibrated confidence estiamtes. Without requiring any architectural changes, our method consistently outperformed the latest loss functions in the domains of calibration and ordinal regression. This work sets a new benchmark for reliable ordinal classification and points the way for future research to optimize regularization parameter $t$ and extend the approach to more diverse and larger datasets, fostering the deployment of more robust calibration methods for a wider range of tasks.

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

## A  APPENDIX

You may include other additional sections here.

