# OpenReview forum: "Calibration of ordinal regression networks"
_ICLR.cc/2025/Conference — ICLR 2025 Conference Withdrawn Submission_

### Official Review · Reviewer_kYWf · 2024-10-30

**Soundness:** 1
**Presentation:** 1
**Contribution:** 2
**Rating:** 3
**Confidence:** 5

**Summary:**

The authors proposed an addition to the cross entropy loss term that  corrects overconfident predictions for incorrect labels. The advantage of the proposed loss is that the calibration is done jointly with the accurate prediction learning during the training optimization and doesn’t require additional post-training steps. The method was evaluated on the 3 datasets and compared against CE-based ordinal models and calibration-loss based and showed an improvement in calibration/accuracy metrics.

**Strengths:**

- This work addresses an important overconfidence issue in ordinal regression tasks
- The proposed loss function is assumed to address both accuracy and confidence of the cross entropy loss based model during optimization without additional post-training calibration
- The authors justify the unimodality enforcement of the proposed loss by gradient analysis

**Weaknesses:**

**Major**:
- The main focus of the work is CE-loss based ordinal regression which is not an optimal loss for this task and  several methods were proposed without CE loss:  [1-4]
- The motivation in Sec 3.1 is unclear, how the calibration is defined and why it is not implied by CE-loss. The discussion seems to be valid for the ordered nature of classes but not for calibration. It is better to discuss the motivation for each problem separately.
 - $\mathcal{L}_{SCE}$ - the explanation in L175-177 is unclear, how the defined loss encourages what the authors claim - maybe it is explained by Diaz et. al but the manuscript should be self contained with additional clarification. It is also not clear how it helps to reflect the ordinal relationships.
- The Sec. 3.3 in unclear, the explanation and derivation of the loss formula should come before presenting the loss term
  - why the authors choose it
  - how it helps to ensure calibration
  - what is r ?
  - The L181-183 is unclear.

- Weak evaluation with only 3 small-sample datasets - overall the improvement is incremental so presenting results on more datasets could be beneficial.
- Missing unimodality evaluation - the authors claim the model enforces unimodality - please show it in the results as in [5]
- Missing additional deep ordinal regression baselines [1-5].
- Sec. 3.4 - while it is clear why the loss term enforces unimodality, I’m not sure how it enforces calibration. By saying “*by increasing the gradient for such incorrect predictions, the model is able to reduce the predicted probability for the incorrect class more effectively*” you can claim the same for the standard CE loss.
- while it could be seen from the results that calibration metrics improved, I’m not sure it is clear from the manuscript why it works.

**Minor**:
- Missing additional deep ordinal regression methods in the related work discussion
- It is better to put Figure 1 closer to the gradient analysis section to make it easier to follow

References:
[1] Liu, X., et al. (2019a). Unimodal-uniform constrained wasserstein training for medical diagnosis. In Proceedings of the IEEE International Conference on Computer Vision Workshops

[2] Beckham, C. et al.  (2017). Unimodal probability distributions for deep ordinal classification.

[3] Wenzhi Cao et. al (2020). Rank Consistent Ordinal Regression for Neural Networks with Application to Age Estimation. Pattern Recognition

[4] Xintong Shi et al. (2021). Deep Neural Networks for Rank-Consistent Ordinal Regression Based On Conditional Probabilities.

[5] Cardoso, J. S. et. al (2023). Unimodal distributions for ordinal regression

**Questions:**

- Please address the points I raised in the weaknesses part
- The methods that are not based on CE loss - like optimal transport loss - how are the limitations applied to them?
- Missing definition of  z_{n.k} and intermediate step to get f’=y - p, also r is not defined before it presented
- Can you elaborate please why the gradient encourages the model to distribute probability across adjacent classes ( as you claim in the sentence in Line 176) compared to the statement in Lines 147-149

---

### Official Review · Reviewer_vAiU · 2024-10-31

**Soundness:** 2
**Presentation:** 2
**Contribution:** 2
**Rating:** 3
**Confidence:** 4

**Summary:**

The proposed method introduces ORCU, a novel loss function designed to ensure calibration and unimodality in ordinal regression tasks. ORCU leverages soft ordinal encoding and order-aware regularization to produce calibrated and unimodal probability distributions, which are particularly valuable in high-stakes applications requiring reliable confidence estimates and accurate predictions.

**Strengths:**

•	The motivation behind the new method is well-articulated, clearly highlighting the limitations of traditional cross-entropy (CE) loss in ordinal tasks and the miscalibration in modern ordinal regression models.
•	The proposed method is straightforward, and easy to implement.

**Weaknesses:**

1.	It is unclear how the regularization component of ORCU promotes calibration. Lines 240-252 discuss scenarios where the model is under or overconfident, yet this confidence is based on a soft-encoded distribution not directly related to the data, which raises questions about its reflection of "real" confidence. Additionally, I would appreciate a more rigorous explanation of how this regularization approach aligns with the standard mathematical definition of calibration. Could the authors provide a clearer mathematical justification for this relationship?

2.	The paper claims that CE loss leads to overconfident predictions, yet the reliability diagrams presented indicate underconfident outcomes in the experiments, seemingly contradicting this claim. Temperature scaling, a prominent calibration technique, relies on CE, further challenging the assertion that CE is fundamentally flawed for calibration. Could the authors address this discrepancy and clarify why their results show underconfidence in CE where overconfidence might be expected? Additionally, a nuanced discussion of CE’s strengths and limitations for calibration, especially in light of techniques like temperature scaling, would be valuable.

3.	The evaluation is limited to loss function baselines. Including additional non-loss-based methods for ordinal regression, such as the approach presented in https://arxiv.org/pdf/2303.04547 for unimodality could highlight the unique benefits of ORCU more effectively. I recommend incorporating a discussion on why ORCU and loss function-based methods may offer advantages over such approaches.

4.	The experiments were conducted on only three datasets, which limits the scope for evaluating the method’s robustness across a wider range of ordinal regression tasks. Incorporating a more extensive dataset selection would allow for a better assessment of the generalizability of the approach.

**Questions:**

See weaknesses.

---

### Official Review · Reviewer_xcrH · 2024-11-01

**Soundness:** 3
**Presentation:** 2
**Contribution:** 2
**Rating:** 5
**Confidence:** 4

**Summary:**

In this paper, the authors aim to enhance the confidence calibration of ordinal regression in the training stage. The main challenge of this task is to consider the calibration and unimodality together. To address this challenge, they propose a new loss function for ordinal regression, which combines order-aware calibration with a unimodal regularization term (based on the SORD encoding). In particular, their method enforces both calibration and unimodality by explicitly modeling the ordinal relationships between classes. The effectiveness of their method is validated on three public datasets.

**Strengths:**

1. The problem of calibration in the context of ordinal regression sounds novel and important. As far as I know, this work should be the first work to solve this issue.

2. The improvement is significant empirically. From Table 2, we can observe a great improvement in the calibration of ordinal regression models and the classification accuracy is preserved.

**Weaknesses:**

1. The L_{REG} defined in Equation 2 is not clearly explained. In particular, the design of I(r) is hard to understand for readers. It would be better if the authors could elaborate on how the regularization is constructed.

2. The writing of the gradient analysis in Subsection 3.4 is not clear.  The authors may need to improve the writing in this part, or it might be too challenging for readers to follow.

3. The technical novelty of the proposed method is not presented. While the authors claim that the method considers the unimodality compared to current calibration methods and considers the calibration when compared to current Ordinal losses, I am not clear about if this method is newly designed in each aspect. In other words, the authors may need to show the new insight of calibration part compared to calibration methods.

typos:
1. Line 49, Oridnal -> Ordinal.

The major issue of this work is on the writing: readers cannot easily understand why we should design such a regularization and how t works here. I will improve my score if the authors can make it clear in the revised version.

**Questions:**

1. In equation 3, we generally adopt a hyperparameter $\lambda$ to balance the loss and the regularization, like $L = L_1 + \lambda L_2$. Could you explain why this method does not require such a hyperparameter? And why $t$ can control the strength of this regularization?

2. Could you explain why SCE and ACE are improved a lot, but ECE is not?

---

### Official Review · Reviewer_aBHc · 2024-11-05

**Soundness:** 2
**Presentation:** 2
**Contribution:** 2
**Rating:** 5
**Confidence:** 5

**Summary:**

In this paper, authors propose an approach for calibration of ordinal regression. They propose a loss function that introduces order-aware calibration They use soft ordinal encoding and label-smoothing-based regularization to enforce both calibration and unimodality. To show the efficiency of the proposed approach, authors propose extensive experimental results on benchmark datasets.

**Strengths:**

1. a novel regularization term is used to promote unimodularity.
2. Paper is well written and easy to read.

**Weaknesses:**

1. Theoretical proofs of calibration and unimodularity are missing.

**Questions:**

1. Many things in the paper are not defined properly. Please clearly specify the definitions of calibration and modularity.

2. Why MAE is not used as a metric in the experiments, as all the loss functions used are surrogates for it. It would be interesting to see comparison results on MAE.

3. Please explain the regularization term in detail as it is still unclear how it promotes unimodularity. A graphical explanation will also help.

4. A theoretical analysis of calibration and unimodularity of the proposed approach is missing.

5. Is the proposed approach rank consistent? If not, then how frequently it violates ranking of thresholds. An experimental study on it will be helpful.

6. How does the performance of the model vary  with the variation in thevalue of $t$ used in the regularization term.

---

### Note · Authors · 2024-11-13

**Comment:**

We would like to thank our sincere gratitude to the reviewers and Area Chairs for their invaluable feedback and insights. After careful consideration, we have decided to withdraw our submission to further develop and refine our paper based on these constructive comments. Thank you once again for your time and thoughtful evaluation.

**Withdrawal Confirmation:**

I have read and agree with the venue's withdrawal policy on behalf of myself and my co-authors.